# Sparsity Meets Robustness: Channel Pruning for the Feynman-Kac Formalism Principled Robust Deep Neural Nets

## Abstract

Deep neural nets (DNNs) compression is crucial for adaptation to mobile devices. Though many successful algorithms exist to compress naturally trained DNNs, developing efficient and stable compression algorithms for robustly trained DNNs remains widely open. In this paper, we focus on a co-design of efficient DNN compression algorithms and sparse neural architectures for robust and accurate deep learning. Such a co-design enables us to advance the goal of accommodating both sparsity and robustness. With this objective in mind, we leverage the relaxed augmented Lagrangian based algorithms to prune the weights of adversarially trained DNNs, at both structured and unstructured levels. Using a Feynman-Kac formalism principled robust and sparse DNNs, we can at least double the channel sparsity of the adversarially trained ResNet20 for CIFAR10 classification, meanwhile, improve the natural accuracy by $8.69\%$ and the robust accuracy under the benchmark 20 iterations of IFGSM attack by $5.42\%$.

## 1 Introduction

Robust deep neural nets (DNNs) compression is a fundamental problem for secure AI applications in resource-constrained environments such as biometric verification and facial login on mobile devices, and computer vision tasks for the internet of things (IoT) (Cheng et al., 2017; Yao et al., 2017; Mohammadi et al., 2018). Though compression and robustness have been separately addressed in recent years, much less is studied when both players are present and must be satisfied.

To date, many successful techniques have been developed to compress naturally trained DNNs, including neural architecture re-design or searching (Howard et al., 2017; Zhang et al., 2018b), pruning including structured (weights sparsification) (Han et al., 2015; Srinivas & Babu, 2015) and unstructured (channel-, filter-, layer-wise sparsification) (Yang et al., 2019; He et al., 2017), quantization (Zhou et al., 2017; Yin et al., 2019; Courbariaux et al., 2016), low-rank approximation (Denil et al., 2013), knowledge distillation (Polino et al., 2018), and many more (Alvarez & Salzmann, 2016; Liu et al., 2015).

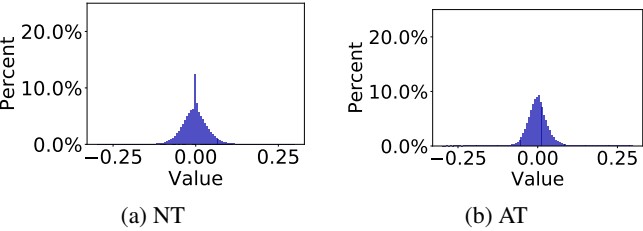

(a) NT          (b) AT

Figure 1: Histograms of the ResNet20's weights.

The adversarially trained (AT) DNN is more robust than the naturally trained (NT) DNN to adversarial attacks (Madry et al., 2018; Athalye et al., 2018). However, adversarial training (denoted as AT if no ambiguity arises, and the same for NT) also dramatically reduces the sparsity of the trained DNN's weights. As shown in Fig. 1, start from the same default initialization in PyTorch, the NT ResNet20's weights are much sparser than that of the AT counterpart, for instance, the percent of

weights that have magnitude less than $10^{-3}$ for NT and AT ResNet20 are 8.66% and 3.64% (averaged over 10 trials), resp. This observation motivates us to consider the following two questions:

- *1. Can we re-design the neural architecture with minimal change on top of the existing one such that the new DNN has sparser weights and better robustness and accuracy than the existing one?*

- *2. Can we develop efficient compression algorithms to compress the AT DNNs with minimal robustness and accuracy degradations?*

We note that under the AT, the recently proposed Feynman-Kac formalism principled ResNet ensemble (Wang et al., 2019a) has much sparser weights than the standard ResNet, which gives a natural answer to the first question above. To answer the second question, we leverage state-of-the-art relaxed augmented Lagrangian based sparsification algorithms (Dinh & Xin, 2018; Yang et al., 2019) to perform both structured and unstructured pruning for the AT DNNs. We focus on unstructured and channel pruning in this work.

## 1.1 NOTATION

Throughout this paper we use bold upper-case letters $\mathbf{A}$, $\mathbf{B}$ to denote matrices, bold lower-case letters $\boldsymbol{x}$, $\boldsymbol{y}$ to denote vectors, and lower case letters $x$, $y$ and $\alpha$, $\beta$ to denote scalars. For vector $\boldsymbol{x} = (x_1, \ldots, x_d)^\top$, we use $\|\boldsymbol{x}\| = \|\boldsymbol{x}\|_2 = \sqrt{x_1^2 + \cdots + x_d^2}$ to represent its $\ell_2$-norm; $\|\boldsymbol{x}\|_1 = \sum_{i=1}^d |x_i|$ to represent its $\ell_1$-norm; and $\|\boldsymbol{x}\|_0 = \sum_{i=1}^d \chi_{\{x_i \neq 0\}}$ to represent its $\ell_0$-norm. For a function $f : \mathbb{R}^d \to \mathbb{R}$, we use $\nabla f(\cdot)$ to denote its gradient. Generally, $\boldsymbol{w}^t$ represents the set of all parameters of the network being discussed at iteration $t$, e.g. $\boldsymbol{w}^t = (\boldsymbol{w}_1^t, \boldsymbol{w}_2^t, ..., \boldsymbol{w}_M^t)$, where $\boldsymbol{w}_j^t$ is the weight on the $j^{th}$ layer of the network at the $t$-th iteration. Similarly, $\boldsymbol{u}^t = (\boldsymbol{u}_1^t, \boldsymbol{u}_2^t, ..., \boldsymbol{u}_M^t)$ is a set of weights with the same dimension as $\boldsymbol{w}^t$, whose value depends on $\boldsymbol{w}^t$ and will be defined below. We use $\mathcal{N}(\mathbf{0}, \mathbf{I}_{d \times d})$ to represent the $d$-dimensional Gaussian, and use notation $O(\cdot)$ to hide only absolute constants which do not depend on any problem parameter.

## 1.2 ORGANIZATION

This paper is organized in the following way: In section 2, we list the most related work to this paper. In section 3, we show that the weights of the recently proposed Feynman-Kac formalism principled ResNet ensemble are much sparser than that of the baseline ResNet, providing greater efficiency for compression. In section 4, we present relaxed augmented Lagrangian-based algorithms along with theoretical analysis for both unstructured and channeling pruning of AT DNNs. The numerical results are presented in section 5, followed by concluding remarks. Technical proofs and more related results are provided in the appendix.

## 2 RELATED WORK

**Compression of AT DNNs:** Gui et al. (2019) considered a low-rank form of the DNN weight matrix with $\ell_0$ constraints on the matrix factors in the AT setting. Their training algorithm is a projected gradient descent (PGD) (Madry et al., 2018) based on the worst adversary. In their paper, the sparsity in matrix factors are unstructured and require additional memory.

**Sparsity and Robustness:** Guo et al. (2018) shows that there is a relationship between the sparsity of weights in the DNN and its adversarial robustness. They showed that under certain conditions, sparsity can improve the DNN's adversarial robustness. The connection between sparsity and robustness has also been studied recently by Ye et al. (2019), Rakin et al. (2019), and et al. In our paper, we focus on designing efficient pruning algorithms integrated with sparse neural architectures to advance DNNs' sparsity, accuracy, and robustness.

**Feynman-Kac formalism principled Robust DNNs:** Neural ordinary differential equations (ODEs) (Chen et al., 2018) are a class of DNNs that use an ODE to describe the data flow of each input data. Instead of focusing on modeling the data flow of each individual input data, Wang et al. (2019a; 2018a); Li & Shi (2017) use a transport equation (TE) to model the flow for the whole input distribution. In particular, from the TE viewpoint, Wang et al. (2019a) modeled training ResNet (He

et al., 2016) as finding the optimal control of the following TE

$$
\begin{cases}
\frac{\partial u}{\partial t}(\boldsymbol{x}, t) + G(\boldsymbol{x}, \boldsymbol{w}(t)) \cdot \nabla u(\boldsymbol{x}, t) = 0, & \boldsymbol{x} \in \mathbb{R}^d, \\
u(\boldsymbol{x}, 1) = g(\boldsymbol{x}), & \boldsymbol{x} \in \mathbb{R}^d, \\
u(\boldsymbol{x}_i, 0) = y_i, & \boldsymbol{x}_i \in T, \text{ with } T \text{ being the training set.}
\end{cases}
\tag{1}
$$

where $G(\boldsymbol{x}, \boldsymbol{w}(t))$ encodes the architecture and weights of the underlying ResNet, $u(\boldsymbol{x}, 0)$ serves as the classifier, $g(\boldsymbol{x})$ is the output activation of ResNet, and $y_i$ is the label of $\boldsymbol{x}_i$.

Wang et al. (2019a) interpreted adversarial vulnerability of ResNet as arising from the irregularity of $u(\boldsymbol{x}, 0)$ of the above TE. To enhance $u(\boldsymbol{x}, 0)$'s regularity, they added a diffusion term, $\frac{1}{2}\sigma^2 \Delta u(\boldsymbol{x}, t)$, to the governing equation of (1) which resulting in the convection-diffusion equation (CDE). By the Feynman-Kac formula, $u(\boldsymbol{x}, 0)$ of the CDE can be approximated by the following two steps:

- Modify ResNet by injecting Gaussian noise to each residual mapping.
- Average the output of $n$ jointly trained modified ResNets, and denote it as $\text{En}_n\text{ResNet}$.

Wang et al. (2019a) have noticed that EnResNet can improve both natural and robust accuracies of the AT DNNs. In this work, we leverage the sparsity advantage of EnResNet to push the sparsity limit of the AT DNNs.

## 3 REGULARITY AND SPARSITY OF THE FEYNMAN-KAC FORMALISM PRINCIPLED ROBUST DNNs' WEIGHTS

From a partial differential equation (PDE) viewpoint, a diffusion term to the governing equation (1) not only smooths $u(\boldsymbol{x}, 0)$, but can also enhance regularity of the velocity field $G(\boldsymbol{x}, \boldsymbol{w}(t))$ (Ladyženskaja et al., 1988). As a DNN counterpart, we expect that when we plot the weights of EnResNet and ResNet at a randomly select layer, the pattern of the former one will look smoother than the latter one. To validate this, we follow the same AT with the same parameters as that used in (Wang et al., 2019a) to train $\text{En}_5\text{ResNet20}$ and ResNet20, resp. After the above two robust models are trained, we randomly select and plot the weights of a convolutional layer of ResNet20 whose shape is $64 \times 64 \times 3 \times 3$ and plot the weights at the same layer of the first ResNet20 in $\text{En}_5\text{ResNet20}$. As shown in Fig. 2 (a) and (b), most of $\text{En}_5\text{ResNet20}$'s weights are close to 0 and they are more regularly distributed in the sense that the neighboring weights are closer to each other than ResNet20's weights. The complete visualization of this randomly selected layer's weights is shown in the appendix. As shown in Fig. 2 (c) and (d), the weights of $\text{En}_5\text{ResNet20}$ are more concentrated at zero than that of ResNet20, and most of the $\text{En}_5\text{ResNet20}$'s weights are close to zero.

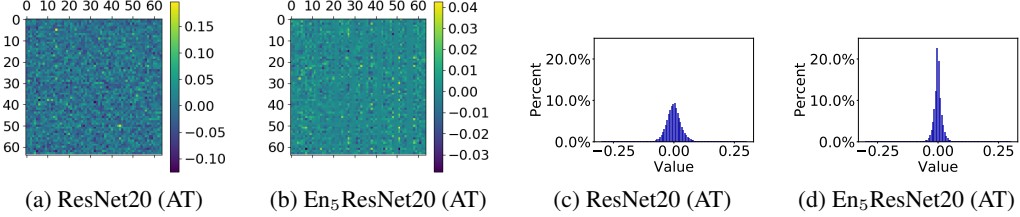

| (a) ResNet20 (AT) | (b) En$_5$ResNet20 (AT) | (c) ResNet20 (AT) | (d) En$_5$ResNet20 (AT) |

Figure 2: (a) and (b): weights visualization; (c) and (d): histogram of weights.

## 4 UNSTRUCTURED AND CHANNEL PRUNING WITH AT

### 4.1 ALGORITHMS

In this subsection, we introduce relaxed, augmented Lagrangian-based, pruning algorithms to sparsify the AT DNNs. The algorithms of interest are the Relaxed Variable-Splitting Method (RVSM) (Dinh & Xin, 2018) for weight pruning (Algorithm 1), and its variation, the Relaxed Group-wise Splitting Method (RGSM) (Yang et al., 2019) for channel pruning (Algorithm 2).

Our approach is to apply the RVSM/RGSM algorithm together with robust PGD training to train and sparsify the model from scratch. Namely, at each iteration, we apply a PGD attack to generate

adversarial images $\boldsymbol{x}'$, which are then used in the forward-propagation process to generate preditions $y'$. The back-propagation process will then compute the appropriate loss function and apply RVSM/RGSM to update the model. Previous work on RVSM mainly focused on a one-hidden layer setting; In this paper, we extend this result to the general setting. To the best of our knowledge, this is the first result that uses RVSM/RGSM in an adversarial training scenario.

To explain our choice of algorithm, we discuss a classical algorithm to promote sparsity of the target weights, the alternating direction of multiplier method (ADMM) (Boyd et al., 2011; Goldstein & Osher, 2009). In ADMM, instead of minimizing the original loss function $f(\boldsymbol{w})$, we seek to minimize the $\ell_1$ regularized loss function, $f(\boldsymbol{w}) + \lambda \|\boldsymbol{u}\|_1$, by considering the following augmented Lagrangian

$$\mathcal{L}(\boldsymbol{w}, \boldsymbol{u}, \boldsymbol{z}) = f(\boldsymbol{w}) + \lambda \|\boldsymbol{u}\|_1 + \langle \boldsymbol{z}, \boldsymbol{w} - \boldsymbol{u} \rangle + \frac{\beta}{2} \|\boldsymbol{w} - \boldsymbol{u}\|^2, \ \lambda, \beta \geq 0. \tag{2}$$

which can be easily solved by applying the following iterations

$$\begin{cases} \boldsymbol{w}^{t+1} \leftarrow \arg\min_{\boldsymbol{w}} \mathcal{L}_{\beta}(\boldsymbol{w}, \boldsymbol{u}^t, \boldsymbol{z}^t) \\ \boldsymbol{u}^{t+1} \leftarrow \arg\min_{\boldsymbol{u}} \mathcal{L}_{\beta}(\boldsymbol{w}^{t+1}, \boldsymbol{u}, \boldsymbol{z}^t) \\ \boldsymbol{z}^{t+1} \leftarrow \boldsymbol{z}^t + \beta(\boldsymbol{w}^{t+1} - \boldsymbol{u}^{t+1}) \end{cases} \tag{3}$$

Although widely used in practice, ADMM has several drawbacks when it is used to regularize DNN's weights. First, one can improve the sparsity of the final learned weights by replacing $\|\boldsymbol{u}\|_1$ with $\|\boldsymbol{u}\|_0$; but $\|\cdot\|_0$ is not differentiable, thus current theory of optimization does not apply (Wang et al., 2018b). Second, the update $\boldsymbol{w}^{t+1} \leftarrow \arg\min_{\boldsymbol{w}} \mathcal{L}_{\beta}(\boldsymbol{w}, \boldsymbol{u}^t, \boldsymbol{z}^t)$ is not a reasonable step in practice, as one has to fully know how $f(\boldsymbol{w})$ behaves. In most ADMM adaptation on DNN, this step is replaced by a simple gradient descent. Third, the Lagrange multiplier term, $\langle \boldsymbol{z}, \boldsymbol{w} - \boldsymbol{u} \rangle$, seeks to close the gap between $\boldsymbol{w}^t$ and $\boldsymbol{u}^t$, and this in turn reduces sparsity of $\boldsymbol{u}^t$.

The RVSM we will implement is a relaxation of ADMM. RVSM drops the Lagrangian multiplier, and replaces $\lambda\|\boldsymbol{u}\|_1$ with $\lambda\|\boldsymbol{u}\|_0$, and resulting in the following relaxed augmented Lagrangian

$$\mathcal{L}_{\beta}(\boldsymbol{w}, \boldsymbol{u}) = f(\boldsymbol{w}) + \lambda \|\boldsymbol{u}\|_0 + \frac{\beta}{2} \|\boldsymbol{w} - \boldsymbol{u}\|^2. \tag{4}$$

The above relaxed augmented Lagrangian can be solved efficiently by the iteration in Algorithm 1. RVSM can resolve all the three issues associated with ADMM listed above in training robust DNNs with sparse weights: First, by removing the linear term $\langle \boldsymbol{z}, \boldsymbol{w} - \boldsymbol{u} \rangle$, one has a closed form formula for the update of $\boldsymbol{u}^t$ without requiring $\|\boldsymbol{u}\|_0$ to be differentiable. Explicitly, $\boldsymbol{u}^t = H_{\sqrt{2\lambda/\beta}}(\boldsymbol{w}^t) = (w_1^t \chi_{\{|w_1| > \sqrt{2\lambda/\beta}\}}, ..., w_d^t \chi_{\{|w_1| > \sqrt{2\lambda/\beta}\}})$, where $H_\alpha(\cdot)$ is the hard-thresholding operator with parameter $\alpha$. Second, the update of $\boldsymbol{w}^t$ is a gradient descent step itself, so the theoretical guarantees will not deviate from practice. Third, without the Lagrange multiplier term $\boldsymbol{z}^t$, there will be a gap between $\boldsymbol{w}^t$ and $\boldsymbol{u}^t$ at the limit (finally trained DNNs). However, the limit of $\boldsymbol{u}^t$ is much sparser than that in the case of ADMM. At the end of each training epoch, we replace $\boldsymbol{w}^t$ by $\boldsymbol{u}^t$ for the validation process. Numerical results in Section 5 will show that the AT DNN with parameters $\boldsymbol{u}^t$ usually outperforms the traditional ADMM in both accuracy and robustness.

---

**Algorithm 1** RVSM

**Input:** $\eta, \beta, \lambda, max_{epoch}, max_{batch}$
**Initialization:** $\boldsymbol{w}^0$
**Define:** $\boldsymbol{u}^0 = H_{\sqrt{2\lambda/\beta}}(\boldsymbol{w}^0)$
**for** $t = 0, 1, 2, ..., max_{epoch}$ **do**
    **for** $batch = 1, 2, ..., max_{batch}$ **do**
        $\boldsymbol{w}^{t+1} \leftarrow \boldsymbol{w}^t - \eta\nabla f(\boldsymbol{w}^t) - \eta\beta(\boldsymbol{w}^t - \boldsymbol{u}^t)$
        $\boldsymbol{u}^{t+1} \leftarrow \arg\min_{\boldsymbol{u}} \mathcal{L}_{\beta}(\boldsymbol{u}, \boldsymbol{w}^t) = H_{\sqrt{2\lambda/\beta}}(\boldsymbol{w}^t)$
    **end for**
**end for**

---

**Algorithm 2** RGSM

**Input:** $\eta, \beta, \lambda_1, \lambda_2, max_{epoch}, max_{batch}$
**Objective:** $\tilde{f}(\boldsymbol{w}) = f(\boldsymbol{w}) + \lambda_2 \|\boldsymbol{w}\|_{GL}$
**Initialization:** Initialize $\boldsymbol{w}^0$, define $\boldsymbol{u}^0$
**for** $g = 1, 2, ..., G$ **do**
    $\boldsymbol{u}_g^0 = Prox_{\lambda_1}(\boldsymbol{w}_g^0)$
**end for**
**for** $t = 0, 1, 2, ..., max_{epoch}$ **do**
    **for** $batch = 1, 2, ..., max_{batch}$ **do**
        $\boldsymbol{w}^{t+1} = \boldsymbol{w}^t - \eta\nabla\tilde{f}(\boldsymbol{w}^t) - \eta\beta(\boldsymbol{w}^t - \boldsymbol{u}^t)$
        **for** $g = 1, 2, ..., G$ **do**
            $\boldsymbol{u}_g^{t+1} = Prox_{\lambda_1}(\boldsymbol{w}_g^t)$
        **end for**
    **end for**
**end for**

---

RGSM is a method that generalizes RVSM to structured pruning, in particular, channel pruning. Let $\boldsymbol{w} = \{\boldsymbol{w}_1, ..., \boldsymbol{w}_g, ..., \boldsymbol{w}_G\}$ be the grouped weights of convolutional layers of a DNN, where $G$ is the total number of groups. Let $I_g$ be the indices of $\boldsymbol{w}$ in group $g$. The group Lasso (GLasso) penalty and group-$\ell_0$ penalty (Yuan & Lin, 2007) are defined as

$$\|\boldsymbol{w}\|_{GL} := \sum_{g=1}^{G} \|\boldsymbol{w}_g\|_2, \qquad \|\boldsymbol{w}\|_{G\ell_0} := \sum_{g=1}^{G} 1_{\|\boldsymbol{w}_g\|_2 \neq 0} \tag{5}$$

and the corresponding Proximal (projection) operators are

$$\text{Prox}_{GL,\lambda}(\boldsymbol{w}_g) := \text{sgn}(\boldsymbol{w}_g)\max(\|\boldsymbol{w}_g\|_2 - \lambda, 0), \qquad \text{Prox}_{G\ell_0,\lambda}(\boldsymbol{w}_g) := \boldsymbol{w}_g 1_{\|\boldsymbol{w}_g\|_2 \neq \sqrt{2\lambda}} \tag{6}$$

where $\text{sgn}(\boldsymbol{w}_g) := \boldsymbol{w}_g / \|\boldsymbol{w}_g\|_2$. The RGSM method is described in Algorithm 2, which improves on adding group Lasso penalty directly in the objective function (Wen et al., 2016) for natural DNN training (Yang et al., 2019).

## 4.2 Theoretical Guarantees

We propose a convergence analysis of the RVSM algorithm to minimize the Lagrangian (4). Consider the following empirical adversarial risk minimization (EARM)

$$\min_{f \in \mathcal{H}} \frac{1}{n} \sum_{i=1}^{n} \max_{\|\boldsymbol{x}_i' - \boldsymbol{x}_i\|_\infty \leq \epsilon} L(F(\boldsymbol{x}_i', \boldsymbol{w}), y_i) \tag{7}$$

where the classifier $F(\cdot, \boldsymbol{w})$ is a function in the hypothesis class $\mathcal{H}$, e.g. ResNet and its ensembles, parametrized by $\boldsymbol{w}$. Here, $L(F(\boldsymbol{x}_i, \boldsymbol{w}), y_i)$ is the appropriate loss function associated with $F$ on the data-label pair $(\boldsymbol{x}_i, y_i)$, e.g. cross-entropy for classification and root mean square error for regression problem. Since our model is trained using PGD AT, let

$$f(\boldsymbol{w}) = \mathbb{E}_{(\boldsymbol{x},y)\sim\mathcal{D}}[\max_{\boldsymbol{x}'} L(F(\boldsymbol{x}', \boldsymbol{w}), y)] \tag{8}$$

where $\boldsymbol{x}'$ is obtained by applying the PGD attack to the clean data $\boldsymbol{x}$ (Wang et al., 2019a; Goodfellow et al., 2014a; Madry et al., 2018; Na et al., 2018). In a nutshell, $f(\boldsymbol{w})$ is the population adversarial loss of the network parameterized by $\boldsymbol{w} = (\boldsymbol{w}_1, \boldsymbol{w}_2, ..., \boldsymbol{w}_M)$. Before proceeding, we first make the following assumption:

**Assumption 1.** *Let $\boldsymbol{w}_1, \boldsymbol{w}_2, ..., \boldsymbol{w}_M$ be the weights in the $M$ layers of the given DNN, then there exists a positive constant $L$ such that for all $t$,*

$$\|\nabla f(\cdot, \boldsymbol{w}_j^{t+1}, \cdot) - \nabla f(\cdot, \boldsymbol{w}_j^t, \cdot)\| \leq L\|\boldsymbol{w}_j^{t+1} - \boldsymbol{w}_j^t\|, \text{ for } j = 1, 2, \cdots, M. \tag{9}$$

Assumption 1 is a weaker version of that made by Wang et al. (2019b); Sinha et al. (2018), in which the empirical adversarial loss function is smooth in both the input $\boldsymbol{x}$ and the parameters $\boldsymbol{w}$. Here we only require the population adversarial loss $f$ to be smooth in each layer of the DNN in the region of iterations. An important consequence of Assumption 1 is

$$f(\cdot, \boldsymbol{w}_j^{t+1}, \cdot) - f(\cdot, \boldsymbol{w}_j^t, \cdot) \leq \langle \nabla f(\cdot, \boldsymbol{w}_j^t, \cdot), (0, ..., \boldsymbol{w}_j^{t+1} - \boldsymbol{w}_j^t, 0, ...)\rangle + \frac{L}{2}\|\boldsymbol{w}_j^{t+1} - \boldsymbol{w}_j^t\|^2 \tag{10}$$

**Theorem 1.** *Under the Assumption 1, suppose also that the RVSM algorithm is initiated with a small stepsize $\eta$ such that $\eta < \frac{2}{\beta+L}$. Then the Lagrangian $\mathcal{L}_\beta(\boldsymbol{w}^t, \boldsymbol{u}^t)$ decreases monotonically and converges sub-sequentially to a limit point $(\bar{\boldsymbol{w}}, \bar{\boldsymbol{u}})$.*

The proof of Theorem 1 is provided in the Appendix. From the descent property of $\mathcal{L}_\beta(\boldsymbol{w}^t, \boldsymbol{u}^t)$, classical results from optimization (Nesterov, 2014) can be used to show that after $T = O(1/\epsilon^2)$ iterations, we have $\nabla_{\boldsymbol{w}^t}\mathcal{L}_\beta(\boldsymbol{w}^t, \boldsymbol{u}^t) = O(\epsilon)$, for some $t \in (0, T]$. The term $\|\boldsymbol{u}\|_0$ promotes sparsity and $\frac{\beta}{2}\|\boldsymbol{w} - \boldsymbol{u}\|^2$ helps keep $\boldsymbol{w}$ close to $\boldsymbol{u}$. Since $\boldsymbol{u} = H_{\sqrt{2\lambda/\beta}}(\boldsymbol{w})$, it follows that $\bar{\boldsymbol{w}}$ will have lots of very small (and thus negligible) components. This result justifies the sparsity in the limit $\bar{\boldsymbol{u}}$.

## 5 NUMERICAL RESULTS

In this section, we verify the following advantages of the proposed algorithms:

- RVSM/RGSM is efficient for unstructured/channel-wise pruning for the AT DNNs.
- After pruning by RVSM and RGSM, EnResNet's weights are significantly sparser than the baseline ResNet's, and more accurate in classifying both natural and adversarial images.

These two merits lead to the fact that a synergistic integration of RVSM/RGSM with the Feynman-Kac formula principled EnResNet enables sparsity to meet robustness.

We perform AT by PGD integrated with RVSM, RGSM, or other sparsification algorithms on-the-fly. For all the experiments below, we run 200 epochs of the PGD (10 iterations of the iterative fast gradient sign method (IFGSM[10]) with $\alpha = 2/255$ and $\epsilon = 8/255$, and an initial random perturbation of magnitude $\epsilon$). The initial learning rate of $0.1$ decays by a factor of 10 at the 80th, 120th, and 160th epochs, and the RVSM/RGSM/ADMM sparsification takes place in the back-propagation stage. We split the training data into 45K/5K for training and validation, and the model with the best validation accuracy is used for testing. We test the trained models on the clean images and attack them by FGSM, IFGSM[20], and C&W with the same parameters as that used in (Wang et al., 2019a; Zhang et al., 2019; Madry et al., 2018). We denote the accuracy on the clean images and under the FGSM, IFGSM[20] (Goodfellow et al., 2014b), C&W (Carlini & Wagner, 2016), and NAttack (Li et al., 2019) [1] attacks as $\boldsymbol{A}_1$, $\boldsymbol{A}_2$, $\boldsymbol{A}_3$, $\boldsymbol{A}_4$, and $\boldsymbol{A}_5$, resp. A brief introduction of these attacks is available in the appendix. We use both sparsity and channel sparsity to measure the performance of the pruning algorithms, where the sparsity is defined to be the percentage of zero weights; the channel sparsity is the percentage of channels whose weights' $\ell_2$ norm is less than $1E - 15$.

### 5.1 MODEL COMPRESSION FOR AT RESNET AND ENRESNETS

First, we show that RVSM is efficient to sparsify ResNet and EnResNet. Table 1 shows the accuracies of ResNet20 and En$_2$ResNet20 under the unstructured sparsification with different sparsity controlling parameter $\beta$. We see that after the unstructured pruning by RVSM, En$_2$ResNet20 has much sparser weights than ResNet20. Moreover, the sparsified En$_2$ResNet20 is remarkably more accurate and robust than ResNet20. For instance, when $\beta = 0.5$, En$_2$ResNet20's weights are 16.42% sparser than ResNet20's (56.34% vs. 39.92%). Meanwhile, En$_2$ResNet20 boost the natural and robust accuracies of ResNet20 from 74.08%, 50.64%, 46.67%, and 57.24% to 78.47%, 56.13%, 49.54%, and 65.57%, resp. We perform a few independent trials, and the random effects is small.

Table 1: Accuracy and sparsity of ResNet20 and En$_2$ResNet20 under different attacks and $\beta$, with $\lambda = 1E - 6$. (Unit: %, n/a: do not perform sparsification. Same for all the following tables.)

| | ResNet20 | | | | | En$_2$ResNet20 | | | | |
|---|---|---|---|---|---|---|---|---|---|---|
| $\beta$ | $\boldsymbol{A}_1$ | $\boldsymbol{A}_2$ | $\boldsymbol{A}_3$ | $\boldsymbol{A}_4$ | Sparsity | $\boldsymbol{A}_1$ | $\boldsymbol{A}_2$ | $\boldsymbol{A}_3$ | $\boldsymbol{A}_4$ | Sparsity |
| n/a | 76.07 | 51.24 | 47.25 | 59.30 | 0 | 80.34 | 57.11 | 50.02 | 66.77 | 0 |
| 0.01 | 70.26 | 46.68 | 43.79 | 55.59 | 80.91 | 72.81 | 51.98 | 46.62 | 63.10 | 89.86 |
| 0.1 | 73.45 | 49.48 | 45.79 | 57.72 | 56.88 | 77.78 | 55.48 | 49.26 | 65.56 | 70.55 |
| 0.5 | 74.08 | 50.64 | 46.67 | 57.24 | 39.92 | 78.47 | 56.13 | 49.54 | 65.57 | 56.34 |

Second, we verify the effectiveness of RGSM in channel pruning. We lists the accuracy and channel sparsity of ResNet20, En$_2$ResNet20, and En$_5$ResNet20 in Table 2. Without any sparsification, En$_2$ResNet20 improves the four type of accuracies by 4.27% (76.07% vs. 80.34%), 5.87% (51.24% vs. 57.11%), 2.77% (47.25% vs. 50.02%), and 7.47% (59.30% vs. 66.77%), resp. When we set $\beta = 1$, $\lambda_1 = 5e - 2$, and $\lambda_2 = 1e - 5$, after channel pruning both natural and robust accuracies of ResNet20 and En$_2$ResNet20 remain close to the unsparsified models, but En$_2$ResNet20's weights are 33.48% (41.48% vs. 8%) sparser than that of ResNet20's. When we increase the channel sparsity level by increasing $\lambda_1$ to $1e - 1$, both the accuracy and channel sparsity gaps between ResNet20 and

---

[1] For NAttack, we use the default parameters in `https://github.com/cmhcbb/attackbox`.

En$_2$ResNet20 are enlarged. En$_5$ResNet20 can future improve both natural and robust accuracies on top of En$_2$ResNet20. For instance, at $\sim 55\%$ (53.36% vs. 56.74%) channel sparsity, En$_5$ResNet20 can improve the four types of accuracy of En$_2$ResNet20 by 4.66% (80.53% vs. 75.87%), 2.73% (57.38% vs. 54.65%), 2.86% (50.63% vs. 47.77%), and 1.11% (66.52% vs. 65.41%), resp.

Table 2: Accuracy and sparsity of different EnResNet20. (Ch. Sp.: Channel Sparsity)

| Net | $\beta$ | $\lambda_1$ | $\lambda_2$ | $A_1$ | $A_2$ | $A_3$ | $A_4$ | $A_5$ | Ch. Sp. |
|---|---|---|---|---|---|---|---|---|---|
| ResNet20 | n/a | n/a | n/a | 76.07 | 51.24 | 47.25 | 59.30 | 45.88 | 0 |
|  | 1 | 5.E-02 | 1.E-05 | 75.91 | 51.52 | 47.14 | 58.77 | 45.02 | 8.00 |
|  | 1 | 1.E-01 | 1.E-05 | 71.84 | 48.23 | 45.21 | 57.09 | 43.84 | 25.33 |
| En$_2$ResNet20 | n/a | n/a | n/a | 80.34 | 57.11 | 50.02 | 66.77 | 49.35 | 0 |
|  | 1 | 5.E-02 | 1.E-05 | 78.28 | 56.53 | 49.58 | 66.56 | 49.11 | 41.48 |
|  | 1 | 1.E-01 | 1.E-05 | 75.87 | 54.65 | 47.77 | 65.41 | 46.77 | 56.74 |
| En$_5$ResNet20 | n/a | n/a | n/a | 81.41 | 58.21 | 51.60 | 66.48 | 50.21 | 0 |
|  | 1 | 1.E-02 | 1.E-05 | 81.46 | 58.34 | 51.35 | 66.84 | 50.07 | 19.76 |
|  | 1 | 2.E-02 | 1.E-05 | 80.53 | 57.38 | 50.63 | 66.52 | 48.23 | 53.36 |

Third, we show that an ensemble of small ResNets via the Feynman-Kac formalism performs better than a larger ResNet of roughly the same size in accuracy, robustness, and sparsity. We AT En$_2$ResNet20 ($\sim 0.54$M parameters) and ResNet38 ($\sim 0.56$M parameters) with and without channel pruning. As shown in Table 3, under different sets of parameters, after RGSM pruning, En$_2$ResNet20 always has much more channel sparsity than ResNet38, also much more accurate and robust. For instance, when we set $\beta = 1$, $\lambda_1 = 5e - 2$, and $\lambda_2 = 1e - 5$, the AT ResNet38 and En$_2$ResNet20 with channel pruning have channel sparsity 17.67% and 41.48%, resp. Meanwhile, En$_2$ResNet20 outperforms ResNet38 in the four types of accuracy by 0.36% (78.28% vs. 77.92%), 3.02% (56.53% vs. 53.51%), 0.23% (49.58% vs. 49.35%), and 6.34% (66.56% vs. 60.32%), resp. When we increase $\lambda_1$, the channel sparsity of two nets increase.. As shown in Fig. 3, En$_2$ResNet20's channel sparsity growth much faster than ResNet38's, and we plot the corresponding four types of accuracies of the channel sparsified nets in Fig. 4.

Table 3: Performance of En$_2$ResNet20 and ResNet38 under RVSM.

| Net | $\beta$ | $\lambda_1$ | $\lambda_2$ | $A_1$ | $A_2$ | $A_3$ | $A_4$ | Ch. Sp. |
|---|---|---|---|---|---|---|---|---|
| En$_2$ResNet20 | n/a | n/a | n/a | **80.34** | **57.11** | **50.02** | **66.77** | 0 |
| ResNet38 | n/a | n/a | n/a | 78.03 | 54.09 | 49.81 | 61.72 | 0 |
| En$_2$ResNet20 | 1 | 5.E-02 | 1.E-05 | **78.28** | **56.53** | **49.58** | **66.56** | **41.48** |
| ResNet38 | 1 | 5.E-02 | 1.E-05 | 77.92 | 53.51 | 49.35 | 60.32 | 17.67 |
| En$_2$ResNet20 | 1 | 1.E-01 | 1.E-05 | **76.30** | **54.65** | **47.77** | **65.41** | **56.74** |
| ResNet38 | 1 | 1.E-01 | 1.E-05 | 72.95 | 49.78 | 46.48 | 57.92 | 43.80 |

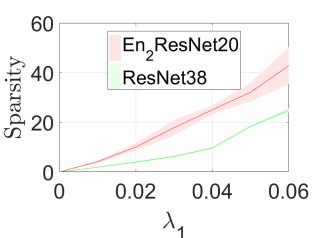

Figure 3: Sparsity of En$_2$ResNet20 and ResNet38 under different parameters $\lambda_1$. (5 runs)

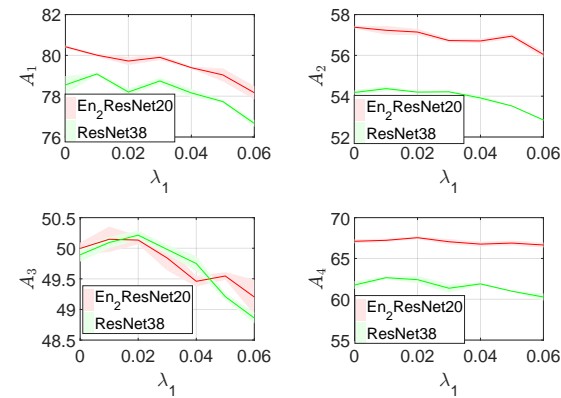

Figure 4: Accuracy of En$_2$ResNet20 and ResNet38 under different parameters $\lambda_1$. (5 runs)

Table 4: Contrasting ADMM versus RVSM for the AT ResNet20.

|  | Unstructured Pruning | | | | | Channel Pruning | | | | |
|---|---|---|---|---|---|---|---|---|---|---|
|  | $A_1$ | $A_2$ | $A_3$ | $A_4$ | Sp. | $A_1$ | $A_2$ | $A_3$ | $A_4$ | Ch. Sp. |
| RVSM | 70.26 | 46.68 | 43.79 | 55.59 | **80.91** | **71.84** | **48.23** | **45.21** | **57.09** | **25.33** |
| ADMM | **71.55** | **47.37** | **44.30** | **55.79** | 10.92 | 63.99 | 42.06 | 39.75 | 51.90 | 4.44 |

## 5.2 RVSM/RGSM VERSUS ADMM

In this subsection, we will compare RVSM, RGSM, and ADMM (Zhang et al., 2018a) [2] for unstructured and channel pruning for the AT ResNet20, and we will show that RVSM and RGSM iterations can promote much higher sparsity with less natural and robust accuracies degradations than ADMM. We list both natural/robust accuracies and sparsities of ResNet20 after ADMM, RVSM, and RGSM pruning in Table 4. For unstructured pruning, ADMM retains slightly better natural ($\sim 1.3\%$) and robust ($\sim 0.7\%$, $\sim 0.5\%$, and $0.2\%$ under FGSM, IFGSM$^{20}$, and C&W attacks) accuracies. However, RVSM gives much better sparsity (80.91% vs. 10.89%). In the channel pruning scenario, RVSM significantly outperforms ADMM in all criterion including natural and robust accuracies and channel sparsity, as the accuracy gets improved by at least $5.19\%$ and boost the channel sparsity from $4.44\%$ to $25.33\%$. Part of the reason for ADMM's inefficiency in sparsifying DNN's weights is due to the fact that the ADMM iterations try to close the gap between the weights $w^t$ and the auxiliary variables $u^t$, so the final result has a lot of weights with small magnitude, but not small enough to be regarded as zero (having norm less than 1e-15). The RVSM does not seek to close this gap, instead it replaces the weight $w^t$ by $u^t$, which is sparse, after each epoch. This results in a much sparser final result, as shown in Figure 5: ADMM does result in a lot of channels with small norms; but to completely prune these off, RVSM does a better job. Here, the channel norm is defined to be the $\ell_2$ norm of the weights in each channel of the DNN (Wen et al., 2016).

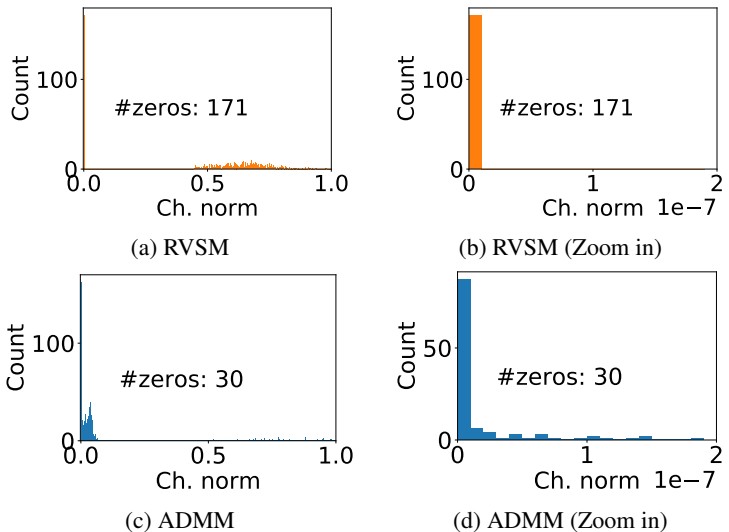

Figure 5: Channel norms of the AT ResNet20 under RVSM and ADMM.

## 5.3 BEYOND RESNET ENSEMBLE AND BEYOND CIFAR10

## 5.4 BEYOND RESNET ENSEMBLE

In this part, we show that the idea of average over noise injected ResNet generalizes to other types of DNNs, that is, the natural and robust accuracies of DNNs can be improved by averaging over noise injections. As a proof of concept, we remove all the skipping connections in ResNet20 and

---

[2]We use the code from: `github.com/KaiqiZhang/admm-pruning`

$\text{En}_2\text{ResNet20}$, in this case, the model no longer involves a TE and CDE. As shown in Table 5, removing the skip connection degrades the performance significantly, especially for $\text{En}_2\text{ResNet20}$ (all the four accuracies of the AT $\text{En}_2\text{ResNet20}$ reduces more than the AT ResNet20's). These results confirm that skip connection is crucial, without it the TE model assumption breaks down, and the utility of EnResNets reduces. However, the ensemble of noise injected DNNs still improves the performance remarkably.

Table 5: Accuracies of ResNet20 and $\text{En}_2\text{ResNet20}$ with and without skip connections.

| Net | Type | $A_1$ | $A_2$ | $A_3$ | $A_4$ |
|---|---|---|---|---|---|
| ResNet20 | Base Net | 76.07 | 51.24 | 47.25 | 59.30 |
|  | Without skip connection | 75.45 | 51.03 | 47.22 | 58.44 |
| $\text{En}_2\text{ResNet20}$ | Base net | 80.34 | 57.11 | 50.02 | 66.77 |
|  | Without skip connection | 79.12 | 55.76 | 49.92 | 66.26 |

Next, let us consider the sparsity, robustness, and accuracies of ResNet20 and $\text{En}_2\text{ResNet20}$ without any skip connection, when they are AT using weights sparsification by either RVSM or RGSM. For RVSM pruning, we set $\beta = 5E - 2$ and $\lambda = 1E - 6$; for RGSM channel pruning, we set $\beta = 5E - 2$, $\lambda_1 = 5E - 2$, and $\lambda_2 = 1E - 5$. We list the corresponding results in Table 6. We see that under both RVSM and RGSM pruning, $\text{En}_2\text{ResNet20}$ is remarkably more accurate on both clean and adversarial images, and significantly sparser than ResNet20. When we compare the results in Table 6 with that in Tables 1 and 2, we conclude that once we remove the skip connections, the sparsity, robustness, and accuracy degrades dramatically.

Table 6: Sparsity and accuracies of ResNet20 and $\text{En}_2\text{ResNet20}$ without skip connection under different pruning algorithms.

| Net | Pruning Algorithm | $A_1$ | $A_2$ | $A_3$ | $A_4$ | Sp. | Ch. Sp. |
|---|---|---|---|---|---|---|---|
| ResNet20 
 (no skip connection) | RVSM | 73.72 | 50.46 | 46.98 | 58.28 | 0.05 | 0.15 |
|  | RGSM | 74.63 | 50.44 | 46.86 | 58.05 | 1.64 | 9.04 |
| $\text{En}_2\text{ResNet20}$ 
 (no skip connection) | RVSM | 76.95 | 55.17 | 49.28 | 58.35 | 10.87 | 9.48 |
|  | RGSM | 78.51 | 56.55 | 49.71 | 67.08 | 7.95 | 15.48 |

## 5.5 BEYOND CIFAR10

Besides CIFAR10, we further show the advantage of EnResNet + RVSM/RGSM in compressing and improving accuracy/robustness of the AT DNNs for CIFAR100 classification. We list the natural and robust accuracies and channel sparsities of the AT ResNet20 and $\text{En}_2\text{ResNet20}$ with different RGSM parameters (n/a stands for do not perform channel pruning) in Table 7. For $\lambda_1 = 0.05$, RGSM almost preserves the performance of the DNNs without channel pruning, while improving channel sparsity by 7.11% for ResNet20, and 16.89% for $\text{En}_2\text{ResNet20}$. As we increase $\lambda_1$ to 0.1, the channel sparsity becomes 18.37% for ResNet20 and 39.23% for $\text{En}_2\text{ResNet20}$. Without any channel pruning, $\text{En}_2\text{ResNet20}$ improves natural accuracy by 4.66% (50.68% vs. 46.02%), and robust accuracies by 5.25% (30.2% vs. 24.77%), 3.02% (26.25% vs. 23.23%), and 7.64% (40.06% vs. 32.42%), resp., under the FGSM, IFGSM[20], and C&W attacks. Even in very high channel sparsity scenario ($\lambda_1 = 0.05$), $\text{En}_2\text{ResNet20}$ still dramatically increase $A_1$, $A_2$, $A_3$, and $A_4$ by 2.90%, 4.31%, 1.89%, and 5.86%, resp. These results are similar to the one obtained on the CIFAR10 in Table 2. These results further confirm that RGSM together with the Feynman-Kac formalism principled ResNets ensemble can significantly improve both natural/robust accuracy and sparsity on top of the baseline ResNets.

## 6 CONCLUDING REMARKS

The Feynman-Kac formalism principled AT EnResNet's weights are much sparser than the baseline ResNet's. Together with the relaxed augmented Lagrangian based unstructured/channel pruning

Table 7: Accuracy and sparsity of different Ensembles of ResNet20's on the CIFAR100.

| Net | $\beta$ | $\lambda_1$ | $\lambda_2$ | $A_1$ | $A_2$ | $A_3$ | $A_4$ | Ch. Sp. |
|---|---|---|---|---|---|---|---|---|
| ResNet20 | n/a | n/a | n/a | 46.02 | 24.77 | 23.23 | 32.42 | 0 |
| | 1 | 5.E-02 | 1.E-05 | 45.74 | 25.34 | 23.55 | 33.53 | 7.11 |
| | 1 | 1.E-01 | 1.E-05 | 44.34 | 24.46 | 23.12 | 32.38 | 18.37 |
| $En_2$ResNet20 | n/a | n/a | n/a | 50.68 | 30.2 | 26.25 | 40.06 | 0 |
| | 1 | 5.E-02 | 1.E-05 | 50.56 | 30.33 | 26.23 | 39.85 | 16.89 |
| | 1 | 1.E-01 | 1.E-05 | 47.24 | 28.77 | 25.01 | 38.24 | 39.23 |

algorithms, we can compress the AT DNNs much more efficiently, meanwhile significantly improves both natural and robust accuracies of the compressed model. As future directions, we propose to quantize EnResNets and to integrate neural ODE into our framework.

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

## A   PROOF OF THEOREM 1

By the $\arg\min$ update of $\boldsymbol{u}^t$, $\mathcal{L}_\beta(\boldsymbol{w}^{t+1}, \boldsymbol{u}^{t+1}) \leq \mathcal{L}_\beta(\boldsymbol{w}^{t+1}, \boldsymbol{u}^t)$. It remains to show $\mathcal{L}_\beta(\boldsymbol{w}^{t+1}, \boldsymbol{u}^t) \leq \mathcal{L}_\beta(\boldsymbol{w}^t, \boldsymbol{u}^t)$. To this end, we show

$$\mathcal{L}_\beta(\boldsymbol{w}_1^{t+1}, \boldsymbol{w}_2^t, ..., \boldsymbol{w}_M^t, \boldsymbol{u}^t) \leq \mathcal{L}_\beta(\boldsymbol{w}_1^t, \boldsymbol{w}_2^t, ..., \boldsymbol{w}_M^t, \boldsymbol{u}^t)$$

the conclusion then follows by a repeated argument. Notice the only change occurs in the first layer $\boldsymbol{w}_1^t$. For simplicity of notation, only for the following chain of inequality, let $\boldsymbol{w}^t = (\boldsymbol{w}_1^t, ..., \boldsymbol{w}_M^t)$ and $\boldsymbol{w}^{t+1} = (\boldsymbol{w}_1^{t+1}, \boldsymbol{w}_2^t, ..., \boldsymbol{w}_M^t)$. Then for a fixed $\boldsymbol{u} := \boldsymbol{u}^t$, we have

$$\begin{aligned}
&\mathcal{L}_\beta(\boldsymbol{w}^{t+1}, \boldsymbol{u}) - \mathcal{L}_\beta(\boldsymbol{w}^t, \boldsymbol{u}) \\
=& f(\boldsymbol{w}^{t+1}) - f(\boldsymbol{w}^t) + \frac{\beta}{2} \left( \|\boldsymbol{w}^{t+1} - \boldsymbol{u}\|^2 - \|\boldsymbol{w}^t - \boldsymbol{u}\|^2 \right) \\
\leq& \langle \nabla f(\boldsymbol{w}^t), \boldsymbol{w}^{t+1} - \boldsymbol{w}^t \rangle + \frac{L}{2} \|\boldsymbol{w}^{t+1} - \boldsymbol{w}^t\|^2 + \frac{\beta}{2} \left( \|\boldsymbol{w}^{t+1} - \boldsymbol{u}\|^2 - \|\boldsymbol{w}^t - \boldsymbol{u}\|^2 \right) \\
=& \frac{1}{\eta} \langle \boldsymbol{w}^t - \boldsymbol{w}^{t+1}, \boldsymbol{w}^{t+1} - \boldsymbol{w}^t \rangle - \beta \langle \boldsymbol{w}^t - \boldsymbol{u}, \boldsymbol{w}^{t+1} - \boldsymbol{w}^t \rangle \\
&+ \frac{L}{2} \|\boldsymbol{w}^{t+1} - \boldsymbol{w}^t\|^2 + \frac{\beta}{2} \left( \|\boldsymbol{w}^{t+1} - \boldsymbol{u}\|^2 - \|\boldsymbol{w}^t - \boldsymbol{u}\|^2 \right) \\
=& \frac{1}{\eta} \langle \boldsymbol{w}^t - \boldsymbol{w}^{t+1}, \boldsymbol{w}^{t+1} - \boldsymbol{w}^t \rangle + \left( \frac{L}{2} + \frac{\beta}{2} \right) \|\boldsymbol{w}^{t+1} - \boldsymbol{w}^t\|^2 \\
&+ \frac{\beta}{2} \|\boldsymbol{w}^{t+1} - \boldsymbol{u}\|^2 - \frac{\beta}{2} \|\boldsymbol{w}^t - \boldsymbol{u}\|^2 - \beta \langle \boldsymbol{w}^t - \boldsymbol{u}, \boldsymbol{w}^{t+1} - \boldsymbol{w}^t \rangle - \frac{\beta}{2} \|\boldsymbol{w}^{t+1} - \boldsymbol{w}^t\|^2 \\
=& \left( \frac{L}{2} + \frac{\beta}{2} - \frac{1}{\eta} \right) \|\boldsymbol{w}^{t+1} - \boldsymbol{w}^t\|^2
\end{aligned}$$

Thus, when $\eta \leq \frac{2}{\beta+L}$, we have $\mathcal{L}_\beta(\boldsymbol{w}^{t+1}, \boldsymbol{u}) \leq \mathcal{L}_\beta(\boldsymbol{w}^t, \boldsymbol{u})$. Apply the above argument repeatedly, we arrive at

$$\mathcal{L}_\beta(\boldsymbol{w}^{t+1}, \boldsymbol{u}^{t+1}) \leq \mathcal{L}_\beta(\boldsymbol{w}^{t+1}, \boldsymbol{u}^t) \leq \mathcal{L}_\beta(\boldsymbol{w}_1^{t+1}, \boldsymbol{w}_2^{t+1}, ..., \boldsymbol{w}_{M-1}^{t+1}, \boldsymbol{w}_M^t, \boldsymbol{u}^t) \leq ... \leq \mathcal{L}_\beta(\boldsymbol{w}^t, \boldsymbol{u}^t)$$

This implies $\mathcal{L}_\beta(\boldsymbol{w}^t, \boldsymbol{u}^t)$ decreases monotonically. Since $\mathcal{L}_\beta(\boldsymbol{w}^t, \boldsymbol{u}^t) \geq 0$, $(\boldsymbol{w}^t, \boldsymbol{u}^t)$ must converge sub-sequentially to a limit point $(\bar{\boldsymbol{w}}, \bar{\boldsymbol{u}})$. This completes the proof.

## B   ADVERSARIAL ATTACKS USED IN THIS WORK

We focus on the $\ell_\infty$ norm based untargeted attack. For a given image-label pair $\{\boldsymbol{x}, y\}$, a given ML model $F(\boldsymbol{x}, \boldsymbol{w})$, and the associated loss $L(\boldsymbol{x}, y) := L(F(\boldsymbol{x}, \boldsymbol{w}), y)$:

- Fast gradient sign method (FGSM) searches an adversarial, $\boldsymbol{x}'$, within an $\ell_\infty$-ball as

$$\boldsymbol{x}' = \boldsymbol{x} + \epsilon \cdot \text{sign} \left( \nabla_{\boldsymbol{x}} L(\boldsymbol{x}, y) \right).$$

- Iterative FGSM (IFGSM$^M$) (Goodfellow et al., 2014b) iterates FGSM and clip the range as

$$\boldsymbol{x}^{(m)} = \text{Clip}_{\boldsymbol{x}, \epsilon} \left\{ \boldsymbol{x}^{(m-1)} + \alpha \cdot \text{sign} \left( \nabla_{\boldsymbol{x}^{(m-1)}} L(\boldsymbol{x}^{(m-1)}, y) \right) \right\}, \text{ w/ } \boldsymbol{x}^{(0)} = \boldsymbol{x}, \ m = 1, \cdots, M.$$

- C&W attack (Carlini & Wagner, 2016) searches the minimal perturbation ($\delta$) attack as

$$\min_\delta \|\delta\|_\infty, \text{ subject to } F(\boldsymbol{w}, \boldsymbol{x} + \delta) = t, \ \boldsymbol{x} + \delta \in [0, 1]^d, \text{ for } \forall t \neq y.$$

- NAttack (Li et al., 2019) is an effective gradient-free attack.

## C    MORE VISUALIZATIONS OF THE DNNS' WEIGHTS

In section 3, we showed some visualization results for part of the weights of a randomly selected convolutional layer of the AT ResNet20 and En$_5$ResNet20. The complete visualization results of this selected layer are shown in Figs. 6 and 7, resp., for ResNet20 and En$_5$ResNet20. These plots further verifies that:

- The magnitude of the weights of the adversarially trained En$_5$ResNet20 is significantly smaller than that of the robustly trained ResNet20.
- The overall pattern of the weights of the adversarially trained En$_5$ResNet20 is more regular than that of the robustly trained ResNet20.

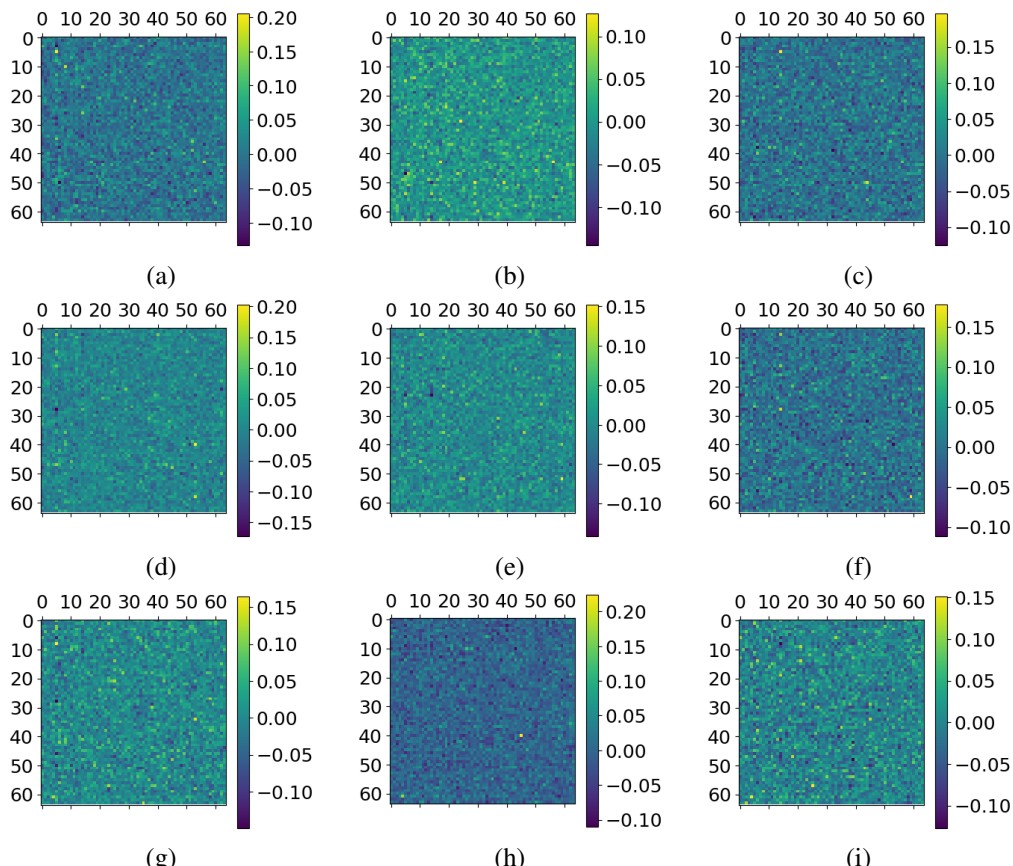

Figure 6: Weights of a randomly selected convolutional layer of the PGD AT ResNet20.

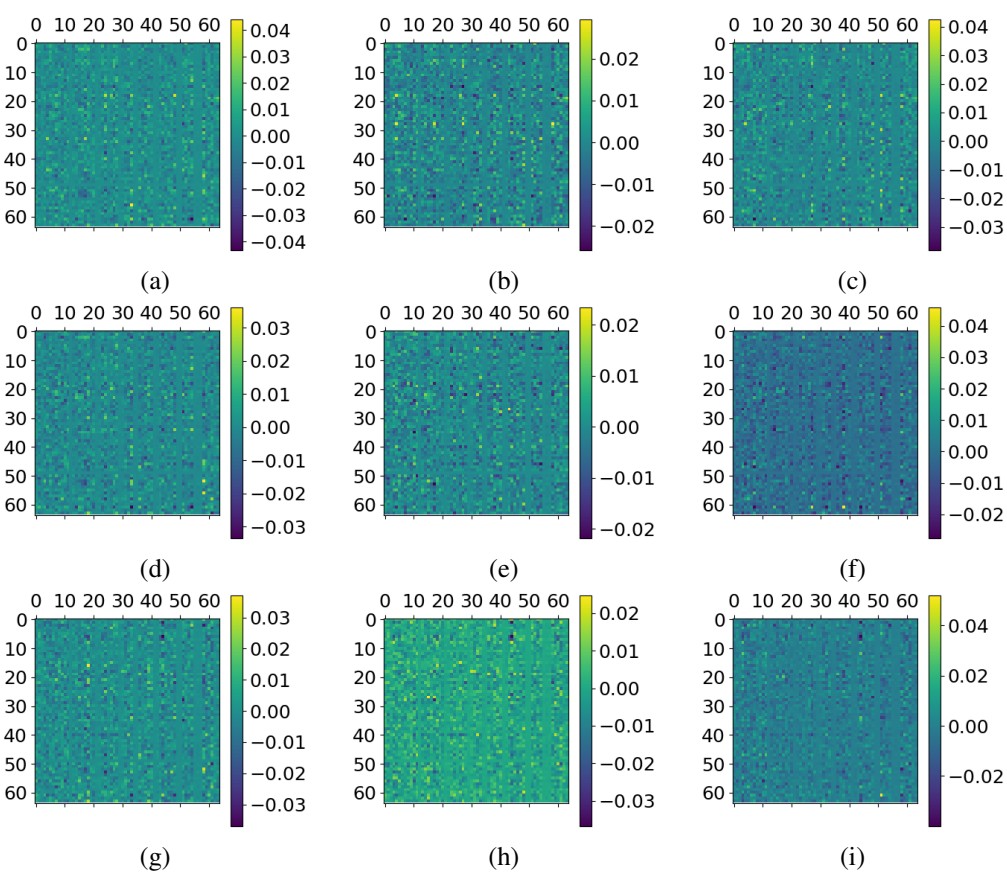

Figure 7: Weights of the PGD AT $En_5$ResNet20 at the same layer as that shown in Fig. 6.

