# OpenReview forum: "Sparsity Meets Robustness: Channel Pruning for the Feynman-Kac Formalism Principled Robust Deep Neural Nets"
_ICLR.cc/2020/Conference — Reject_

### Official Review · AnonReviewer3 · 2019-10-23
**Official Blind Review #3**

**Rating:** 3

**Review:**

Summary:
This paper focuses on the study of sparse neural architectures and efficient DNN compression algorithms for the robust and accurate deep learning. The authors apply relaxed augmented Lagrangian based sparsification algorithms to perform both unstructured and channel pruning for the AT DNNs. By combing with the sparsity merit of EnResNet, their sparsification algorithms further boost the sparsity limit of the AT DNNs, leading to much better robustness and accuracy. Their approach demonstrates superior natural and robust accuracies under several benchmark attacks.

Strengths:
1 The authors practically apply the RVSM/RGSM algorithms to the AT using robust PGD training, resulting in significantly enhanced sparsity of DNNs, and achieving promising robust accuracies against various adversarial attacks.
2 The authors provide the theoretical analysis showing the convergence of the RVSM algorithm.

Weaknesses:
1 The main contribution of paper is to adapt the RVSM/RGSM algorithms to the AT of DNNs to sparsify the deep model. However, both RVSM and RGSM have already been used as sparsification algorithms for DNNs, and the Feynman-Kac formalism principled DNNs has also been investigated for the purpose of sparsification, thus the contribution is largely reduced and the novelty appear to be limited.

2 For the experiments, the effectiveness of RVSM/RGSM are verified on the variants of ResNet and EnResNet, and there are insufficient comparisons with other cutting-edge compression/sparsification methods to show the advantage of the proposed method.

Question:
1 The author claim that ADMM produce a lot of small weights that cannot be regarded as zero since the norm is large than 1e-15. Is the ‘1e-15’ threshold too small to regard a value as zero?

**Experience Assessment:**

I have published one or two papers in this area.

**Review Assessment: Checking Correctness Of Derivations And Theory:**

I did not assess the derivations or theory.

**Review Assessment: Checking Correctness Of Experiments:**

I assessed the sensibility of the experiments.

**Review Assessment: Thoroughness In Paper Reading:**

I made a quick assessment of this paper.

---

> ### Author Response · Authors · 2019-11-12
> **Reply to review #3**
>
> Thank you for your valuable feedback and thoughtful reviews. Below we address your concerns about our paper.
>
> Q1. The main contribution of paper is to adapt the RVSM/RGSM algorithms to the AT of DNNs to sparsify the deep model. However, both RVSM and RGSM have already been used as sparsification algorithms for DNNs, and the Feynman-Kac formalism principled DNNs has also been investigated for the purpose of sparsification, thus the contribution is largely reduced and the novelty appear to be limited.
> Answer: The authors of EnResNet just used the fact that the ensemble of the adversarially trained
> noise injected ResNet can improve both natural and robust accuracy on top of the baseline ResNet. The novelty of our work is twofold: First, based on the property of the convection-diffusion equation, we observed that the weights of the adversarially trained EnResNet are much sparser than the baseline ResNet. Second, we proposed a relaxed augmented Lagrangian based algorithm to efficiently prune the weights at both structured and unstructured level. The relaxed augmented Lagrangian method is more efficient than the standard ADMM approach for pruning the weights of deep neural nets. Moreover, the pruning algorithm provably converges in the adversarial setting. Both the sparsity observation and the relaxed augmented Lagrangian based algorithm for structured and unstructured pruning are novel.
>
>
> Q2. For the experiments, the effectiveness of RVSM/RGSM are verified on the variants of ResNet and EnResNet, and there are insufficient comparisons with other cutting-edge compression/sparsification methods to show the advantage of the proposed method.
> Answer: In this paper, we compared the proposed RVSM/RGSM with ADMM type structured/unstructured pruning algorithms that are also Lagrangian based and used recently by other researchers. We shall compare with other sparsification methods in future work, however, we already made the first major step forward in achieving considerable sparsity for robust networks. Moreover, RVSM/RGSM is simple and efficient, easy to implement and enjoys a solid theoretical foundation without any ad hoc modification.
>
>
>
> Q3. The author claims that ADMM produce a lot of small weights that cannot be regarded as zero since the norm is larger than 1e-15. Is the ‘1e-15’ threshold too small to regard a value as zero?
>
> Answer: A small threshold value 1e-15 is meaningful in that the individual weight values in a channel with norm above such threshold may still exceed machine precision and have a non-negligible effect on accuracy. In the case of 64 bit, the value for machine precision is approximately 2.22e-16. Thus 1.e-15 is a reasonable threshold. One can use a larger threshold, yet it remains true that ADMM does not zero out as many weights (channels) as RVSM (RGSM).
>
> ================================================
> We hope we have answered your questions about our work, and we would appreciate more constructive feedback from you.

---

### Official Review · AnonReviewer1 · 2019-10-24
**Official Blind Review #1**

**Rating:** 3

**Review:**

Summary: This paper presents an observation that Feynman-Kac formalism principled ResNet ensemble [1] yields sparser network weights compared to those of standard Resnet.  Based on this observation, the authors combine the ensemble model with Lagrangian based sparsification methods to obtain sparse and robust models.

Pros:
- Obtaining sparse and robust networks is an important and challenging problem that could be of interest to a large audience.
- The paper provides an interesting observation that EnResNet [1] yields sparser network weights compared to standard ResNet, and further leverage it to obtain sparse and robust models.

Cons:
- The paper has limited novelty. It has already been shown in the original EnResNet paper [1] that EnResNet is more robust to adversarial attacks. Thus the only additional contribution of this paper is the observation that EnResnet is more sparse than standard ResNet, and combination of EnResnet with sparsification methods.
- The paper is not well justified on why one should use sparsifying techniques such as RVSM, RGSM with the Feynman-Kac formula principled EnResnet. The authors state that this enables sparsity to meet robustness; however, in all experimental results, the robustness actually decreases with the increase in sparsity which is opposite to the claims made in the paper.
- The authors only report robustness on white-box gradient-based attacks. Thus it is not clear whether the method will generalize to black-box/gradient-free attack approaches such as NAttack [2].

Minor comments:

- Why didn’t you report the accuracy on the clean examples? This is important in showing that the method generalizes well to clean examples while maintaining robustness.
- Why use only 20 iterations to evaluate the attack? Will the model maintain its robustness with the increase in the iteration and epsilon?
- Figures are not referenced anywhere in the text.
- It would be better to put the results on CIFAR100 in the main paper rather than in the appendix.
- Page 2, “lower cases”  -> “lower case”.
- Page 2, “Related Works” -> “Related work”, it’s better to have it as a separate section.

Overall, while the paper provides an interesting observation, it has limited contributions due to lack of novelty. Further, inadequate experimental validation makes it difficult to see if the claims made in the paper are actually true. Thus I believe that this paper requires substantial improvements in order to be accepted to top-tier publication venues such as ICLR.

References:
[1] Bao et al. 2018, https://arxiv.org/abs/1811.10745
[2] Yadong et al. 2019, https://arxiv.org/abs/1905.00441

**Experience Assessment:**

I have published in this field for several years.

**Review Assessment: Checking Correctness Of Derivations And Theory:**

I assessed the sensibility of the derivations and theory.

**Review Assessment: Checking Correctness Of Experiments:**

I carefully checked the experiments.

**Review Assessment: Thoroughness In Paper Reading:**

I read the paper at least twice and used my best judgement in assessing the paper.

---

> ### Author Response · Authors · 2019-11-12
> **Reply to review #1**
>
> Thank you for your valuable feedback and thoughtful reviews. Below we address your concerns about our paper.
>
> Q1. The paper has limited novelty. It has already been shown in the original EnResNet paper [1] that EnResNet is more robust to adversarial attacks. Thus the only additional contribution of this paper is the observation that EnResnet is more sparse than standard ResNet, and combination of EnResnet with sparsification methods. Overall, while the paper provides an interesting observation, it has limited contributions due to lack of novelty. Further, inadequate experimental validation makes it difficult to see if the claims made in the paper are actually true. Thus I believe that this paper requires substantial improvements in order to be accepted to top-tier publication venues such as ICLR.
> Answer: The adversarially trained EnResNet is observed to be more robust in [1]. The novelty of our work is twofold: First, based on the property of the convection-diffusion equation, we observed that the weights of the adversarially trained EnResNet are much sparser than the baseline ResNet. Second, we proposed a relaxed augmented Lagrangian based algorithm to efficiently prune the weights at both structured and unstructured level. The relaxed augmented Lagrangian method is more efficient than the standard ADMM approach for pruning the weights of deep neural nets. Moreover, the pruning algorithm provably converges in the adversarial setting. Both the sparsity observation and the relaxed augmented Lagrangian based algorithm for structured and unstructured pruning are novel.
>
> In our paper, we conducted extensive experiments on both CIFAR10 and CIFAR100 to verify the robustness and sparsity of the proposed framework. In the revised manuscript, we have also added more results on adversarial attacks.
>
> Q2. The paper is not well justified on why one should use sparsifying techniques such as RVSM, RGSM with the Feynman-Kac formula principled EnResnet. The authors state that this enables sparsity to meet robustness; however, in all experimental results, the robustness actually decreases with the increase in sparsity which is opposite to the claims made in the paper.
> Answer: In this paper, we have two objectives in mind. First, we perform the neural net architecture redesign to improve their sparsity. Second, on top of the new architecture, we develop efficient sparsification algorithms to prune the neural net at both structured and unstructured sparsity levels. We leveraged both RVSM and RGSM to achieve the second goal. In our paper, sparsity meets robustness means that  we are developing algorithms to improve both robustness and sparsity on top of the baseline deep neural nets. We do not mean that improving sparsity will improve robustness.
>
> Q3. The authors only report robustness on white-box gradient-based attacks. Thus it is not clear whether the method will generalize to black-box/gradient-free attack approaches such as NAttack.
> Answer: We list the results on the black-box adversarial attack below, where we use the base model to classify the adversarial examples crafted by attacking the target model. All models are PGD adversarially trained with $\beta=1$, $\lambda_1=0.1$, and $\lambda_2=1e-5$.
> -----------------------------------------------------------------------------------------------------------------------------------------
> Based Model            Target Model       $IFGSM^{10}$  $IFGSM^{20}$  $IFGSM^{50}$   $IFGSM^{100}$
> -----------------------------------------------------------------------------------------------------------------------------------------
> En$_5$ResNet20    ResNet20                   61.63             60.65                 60.67                    60.59
> En$_2$ResNet20    ResNet20                   58.51             56.99                 56.79                    57.01
>     ResNet20         En$_2$ResNet20          57.50             53.54                 53.51                    53.37
>     ResNet20         En$_5$ResNet20          58.14             53.97                 53.87                    53.65
> -----------------------------------------------------------------------------------------------------------------------------------------
>
> In the revised manuscript, we have also added results on NAttack (with the default parameters in the attackbox: https://github.com/cmhcbb/attackbox) in Table 2. Below is a partial list of the results of the three models mentioned above.
> -----------------------------------------------------------
> Model                       Accuracy (NAttack)
> ----------------------------------------------------------
> ResNet20                   43.84
> En$_2$ResNet20          46.77
> En$_5$ResNet20          48.23
> ----------------------------------------------------------
>
> We see that the sparsified EnResNets are also more robust to both black-box IFGSM attack and NAttack than the baseline ResNet.

---

> > ### Author Response · Authors · 2019-11-12
> > **Reply to review #1 -- Continued**
> >
> > Q4. Why didn’t you report the accuracy on the clean examples? This is important in showing that the method generalizes well to clean examples while maintaining robustness.
> > Answer: We call the accuracy on the clean examples as natural accuracy, denoted as $A_1$, in our paper, we followed the name adopted by other researchers. As you expected, our proposed framework not only remarkably improves robustness but also improves their natural accuracy, as shown in Tables 1, 2, 3, and Figure 4.
> >
> > Q5. Why use only 20 iterations to evaluate the attack? Will the model maintain its robustness with the increase in the iteration and epsilon?
> > Answer: 20 iterations IFGSM attack is the commonly used attack to evaluate the robustness of machine learning models. Below we list the accuracy of different models under IFGSM with the different number of iterations. All models are trained with $\beta=1$, $\lambda_1=0.1$, and $\lambda_2=1e-5$.
> > ----------------------------------------------------------------------------------------------------------------
> >        Model            $IFGSM^{10}$  $IFGSM^{20}$  $IFGSM^{50}$   $IFGSM^{100}$
> > ----------------------------------------------------------------------------------------------------------------
> >     ResNet20                48.14             47.14                 46.50                    46.46
> > En$_2$ResNet20        54.46             49.58                 48.65                    48.47
> > En$_5$ResNet20        56.92             51.35                 50.32                    50.19
> > ----------------------------------------------------------------------------------------------------------------
> > It shows that  the robust accuracy of the sparsified EnResNets is significantly higher than ResNet.
> >
> >
> > Q6. Figures are not referenced anywhere in the text.
> > Answer: We referenced the figures in the text. Mostly, we call Fig.~(figure number). Also, we discussed all the figures in the main text.
> >
> > Q7. It would be better to put the results on CIFAR100 in the main paper rather than in the appendix.
> > Answer: As you suggested, we have moved the results on CIFAR100 to the main paper.
> >
> > Q8. Page 2, “lower cases”  -> “lower case”.  Page 2, “Related Works” -> “Related work”, it’s better to have it as a separate section.
> > Answer: As you suggested, we have made the related work part to be a separate section in the revised manuscript and fixed the typos.
> >
> > ================================================
> > We hope we have answered your questions about our work, and we would appreciate more constructive feedback from you.

---

### Official Review · AnonReviewer2 · 2019-10-24
**Official Blind Review #2**

**Rating:** 1

**Review:**

This paper presents an algorithm to train neural networks combining sparsity and adversarial training.
On the sparsity inducing regularization, the paper proposes using proximal methods.

On the positive side:
- The goal is of interest when it comes to real-world applications.
- There is an interesting analysis of the weights and how pruning / AT affects them.


On the negative side:

- In my opinion, the introduction messes unstructured and structured. Weight sparsification is associated with structured while I do believe is unstructured.


- Related work does not seem very comprehensive. The paper claims novelty on using RGSM to improve performance. How is this different from the formulation used in "Learning the Number of Neurons in Deep Networks, Alvarez and Salzmann NIPS 2016".



Experimental settings:
- One thing that got my attention is the threshold for pruning weights (1e-15). I think that is not a fair value. There are related works suggesting no loss in accuracy if the threshold is ~1e-5 (y. Sparse convolutional neural networks CVPR2015). I do believe the numbers would change drastically.

- The comparison between ADMM and the proximal is unfair if using that threshold. The proximal has an implicit thresholding Eq. 6.

- What is the goal for the ensembles of the small networks? where are the numbers?


Minor stuff:
- In sparsity and robustness, there are some works missing (as a typo).
- On the method, there are two parts where I am confused. What is the aim for including all-around Eq. 4? The same with the theoretical guarantees.

**Experience Assessment:**

I have published one or two papers in this area.

**Review Assessment: Checking Correctness Of Derivations And Theory:**

I assessed the sensibility of the derivations and theory.

**Review Assessment: Checking Correctness Of Experiments:**

I assessed the sensibility of the experiments.

**Review Assessment: Thoroughness In Paper Reading:**

I read the paper thoroughly.

---

> ### Author Response · Authors · 2019-11-12
> **Reply to review #2**
>
> Thank you for your valuable feedback and thoughtful reviews. Below we address your concerns about our paper.
>
> Q1. In my opinion, the introduction messes unstructured and structured. Weight sparsification is associated with structured while I do believe is unstructured.
> Answer: In our submission, we focused on the channel-pruning, which is structured pruning. For comparison, we also reported the results on the unstructured pruning — weight sparsification. We have made this more clear in the revised manuscript.
>
> Q2. Related work does not seem very comprehensive. The paper claims novelty on using RGSM to improve performance. How is this different from the formulation used in "Learning the Number of Neurons in Deep Networks, Alvarez and Salzmann NIPS 2016".
> Answer: Alvarez and Salzmann NIPS 2016 is a proximal gradient descent method on group lasso and L1 penalty, while RGSM involves a relaxed variable to explore the loss landscape and sparsify weights. In our paper, RGSM is applied to group L0 (L0) for structure (un-structured) sparsity, which is more effective in realizing sparsity and maintaining accuracy in gradient descent training than using group L1 (L1). To the best of our  knowledge, this is the first time that the L0 penalty is studied in the setting of adversarial learning.
>
> Q3. One thing that got my attention is the threshold for pruning weights (1e-15). I think that is not a fair value. There are related works suggesting no loss in accuracy if the threshold is ~1e-5 (y. Sparse convolutional neural networks CVPR2015). I do believe the numbers would change drastically.
> Answer: The threshold of 1e-15 comes from the observation that the weights in a channel with norm less than 1e-15 is essentially under machine precision and have no effects on accuracy when set to zero.
> Hence, the threshold should be small to keep a good robust accuracy.  We tested ResNet-20 on CIFAR10 with channel norm threshold increased to 1e-5, and found that RVSM has 28% (133 vs. 171) more sparsity than ADMM with comparable or better accuracy.
>
> Q4. The comparison between ADMM and the proximal is unfair if using that threshold. The proximal has an implicit thresholding Eq. 6.
> Answer: Step 2 of ADMM (u update of eq. (3) ) can also be written as a proximal operation with implicit thresholding. Hence the comparison is fair. The difference is that ADMM has an extra multiplier, while RVSM/RGSM does not.
>
> Q5. What is the goal for the ensembles of the small networks? where are the numbers?
> Answer: The goal of the ensemble of noise injected small ResNet is to improve the sparsity and robustness on top of the baseline ResNet. This is one of the crucial components of our proposed framework. We denoted the ensemble of ResNet as En$_x$ResNet, where the subscript x indicates the number of ResNet.
>
> Q6. In sparsity and robustness, there are some works missing (as a typo).
> Answer: They are fixed in the revised manuscript.
>
> Q7. On the method, there are two parts where I am confused. What is the aim for including all-around Eq. 4? The same with the theoretical guarantees.
> Answer: Eq 4 shows the Lagrangian function being minimized by our algorithm. The theoretical results provide a mathematical foundation of our algorithm and its convergence based on Eq 4.
>
> ================================================
> We hope we have answered your questions about our work, and we would appreciate more constructive feedback from you.

---

### Decision · Program_Chairs · 2019-12-19

**Decision:**

Reject

**Comment:**

The paper is rejected based on unanimous reviews.